# Genomic variations of the mevalonate pathway in porokeratosis

Zhenghua Zhang[1]*[†], Caihua Li[2,3][†], Fei Wu[4], Ruixiao Ma[3], Jing Luan[1], Feng Yang[3], Weida Liu[5], Li Wang[3], Shoumin Zhang[6], Yan Liu[3], Jun Gu[7], Wenlian Hua[3], Min Fan[8], Hua Peng[3], Xuemei Meng[9], Ningjing Song[10], Xinling Bi[7], Chaoying Gu[1], Zhen Zhang[1], Qiong Huang[1], Lianjun Chen[1], Leihong Xiang[1], Jinhua Xu[1], Zhizhong Zheng[1], Zhengwen Jiang[3,11]*

[1]Department of Dermatology, Huashan Hospital, Shanghai Medical College of Fudan University, Shanghai, China; [2]School of Life Sciences, Fudan University, Shanghai, China; [3]Genesky Biotechnologies Inc, Shanghai, China; [4]Shanghai Dermatology Hospital, Shanghai, China; [5]Institute of Dermatology, Chinese Academy of Medical Sciences, Nanjing, China; [6]Department of Dermatology, Henan Provincial People's Hospital, Zhengzhou, China; [7]Department of Dermatology, Changhai Hospital, Second Military Medical University, Shanghai, China; [8]Shenzhen Ruimin Dermatology Hospital, Shenzhen, China; [9]Department of Dermatology, Central Hospital of China National Petroleum Corp, Langfang, China; [10]Department of Dermatology, Tongji Hospital, Shanghai Jiaotong University, Shanghai, China; [11]Genesky Diagnostics Inc, BioBay, SIP, Jiangsu, China

*For correspondence:
zhengwenj@geneskies.com (ZJ);
verzhang@foxmail.com (ZZ)

[†]These authors contributed equally to this work

Competing interests: The authors declare that no competing interests exist.

**Abstract** Porokeratosis (PK) is a heterogeneous group of keratinization disorders. No causal genes except *MVK* have been identified, even though the disease was linked to several genomic loci. Here, we performed massively parallel sequencing and exonic CNV screening of 12 isoprenoid genes in 134 index PK patients (61 familial and 73 sporadic) and identified causal mutations in three novel genes (*PMVK*, *MVD*, and *FDPS*) in addition to *MVK* in the mevalonate pathway. Allelic expression imbalance (AEI) assays were performed in 13 lesional tissues. At least one mutation in one of the four genes in the mevalonate pathway was found in 60 (98%) familial and 53 (73%) sporadic patients, which suggests that isoprenoid biosynthesis via the mevalonate pathway may play a role in the pathogenesis of PK. Significantly reduced expression of the wild allele was common in lesional tissues due to gene conversion or some other unknown mechanism. A G-to-A RNA editing was observed in one lesional tissue without AEI. In addition, we observed correlations between the mutations in the four mevalonate pathway genes and clinical manifestations in the PK patients, which might support a new and simplified classification of PK under the guidance of genetic testing.

## Introduction

Porokeratosis (PK, MIM 175800) is a heterogeneous group of keratinization disorders that exhibit an autosomal dominant mode of inheritance. PK is also a skin-specific autoinflammatory disease which was often inherited and linked to ultraviolet light exposure and immunosuppression (*Schamroth et al., 1997*; *Abramovits and Oquendo, 2013*). For example, eruptive pruritic papular porokeratosis exemplifies the inflammatory manifestation, and complications to inflammatory conditions such as localized cutaneous amyloidosis are seen in PK patients (*Biswas, 2015*). Indeed, PK and psoriasis share some features at both clinical and molecular levels and sometimes coexist in the same patients (*Zhang et al., 2008*).

**eLife digest** Porokeratosis refers to a group of around twenty skin conditions that involve a build-up of a protein called keratin in skin cells. Keratin forms the tough fibres that give strength to hair and nails, and people suffering from porokeratosis develop hardened skin lesions. Porokeratosis is an uncommon condition; most cases are inherited and have been linked to exposure to ultraviolet light and having a weakened immune system.

Mutations in one gene called *MVK* are known to cause two forms of the disorder, but it is suspected that other genetic causes of porokeratosis will also be identified. The *MVK* gene encodes an enzyme that is involved in making chemicals called isoprenoids. This large and diverse class of chemicals provides the building blocks for making many other important molecules in all living species. Zhang, Li et al. have now analysed genetic material from 134 different porokeratosis patients to search for mutations in other genes involved in the production of isoprenoids. The patients examined include 61 people with a family history of the disorder, and 73 cases in which the condition seems to be a one-off occurrence.

This search identified mutations in three additional genes (called *PMVK*, *MVD* and *FDPS*) that are all linked to porokeratosis. Further analysis of these three genes and *MVK* revealed that about half of the patients with mutations in the *MVK* gene developed large lesions (that were over 5 centimetres in diameter). However, those with mutations in the other three genes did not develop such large lesions. Mutations in some of the newly identified genes were instead linked to porokeratosis affecting specific areas of the body; for example, *PMVK* and *MVD* mutations are linked to porokeratosis localized to the genitals and around the eyes, respectively. This means that, in the future, doctors might be able to simplify the diagnosis of the different varieties of porokeratosis based on information gained via genetic tests.

As a histological hallmark that unifies all variants of PK, cornoid lamella (CL) is a vertical 'column' of parakeratosis. The pattern of CL can be slender, broad, or confluent, which is related to epidermal hyperplasia and dermal inflammation. However, CL is not a unique feature of PK because it can be seen in some inflammatory and inherited cutaneous disorders and also as an incidental finding (*Biswas, 2015*).

PK is currently classified according to the clinical manifestations, such as number, size, morphology, and distribution of the histological lesions. A better system of classification is expected because some variants of PK are fraught with confusing terminology (*Schamroth et al., 1997*; *Sertznig et al., 2012*; *Biswas, 2015*). For example, it is sometimes hard to completely differentiate disseminated superficial actinic porokeratosis (DSAP) from disseminated superficial porokeratosis (DSP) by age of onset and sun-exposed areas.

In addition to the heterogeneity in clinical manifestations, genetic heterogeneity is also observed in PK. At least five linkage loci (i.e., 12q23.2-24.1, 15q25.1-26.1, 18p11.3, 1p31.3-p31.1, 16q24.1-24.3) have been reported for the disseminated forms of PK which include DSAP, DSP, porokeratosis palmaris et plantaris disseminata (PPPD), and immunosuppression-induced porokeratosis (ISIP) (*Schamroth et al., 1997*; *Luan et al., 2011*). However, only one causal gene, the mevalonate kinase gene (*MVK*) at 12q24, has been identified in DSAP and porokeratosis of Mibille (PM) (*Zhang et al., 2012*; *Li and Zhang, 2014*; *Zeng et al., 2014*). Here we performed genetic analysis in 134 index patients with PK to identify additional causal genes and to establish new parameters for better differential diagnosis of PK. Allelic expression imbalance (AEI) and cDNA mutation were analyzed in 13 pairwise lesional tissues (LTs) and neighboring normal-appearing skin (NNS) to investigate the underlying pathogenicity of PK.

## Results

### Co-segregation of mevalonate (diphospho) decarboxylase (*MVD*) mutation with PK

We previously identified a linkage locus for DSAP on chromosome 16q24.1-24.3 in a four-generation Chinese DSAP family (*Luan et al., 2011*). We applied the candidate gene approach and hypothesized

that the *MVD* gene in the 16q24.1-24.3 region was the causal gene of DSAP in the PK family. We performed Sanger sequencing of all *MVD* exons and identified a c.746T>C (p.Phe249Ser) mutation in *MVD*. The c.746T>C mutation displayed 100% co-segregation with the PK phenotype in this family (*Figure 1*).

## Point mutations in *MVK*, *PMVK*, *MVD*, and *FDPS* were identified in PK patients

Since both *MVD* and *MVK* are involved in isoprenoid biosynthesis via the mevalonate pathway (*Thurnher et al., 2013*), we hypothesized that mutations in other members of the mevalonate pathway exist in PK and screened for mutations in 12 genes in the mevalonate pathway including *MVD* and *MVK* in 134 PK patients (61 familial and 73 sporadic). The 12 genes are: acetyl-CoA acetyltransferase (*ACAT*) *1* and *2*, 3-hydroxy-3-methylglutaryl-CoA synthase (*HMGCS*) *1* and *2*, 3-hydroxy-3-methylglutaryl-CoA reductase (*HMGCR*), *MVK*, phosphomevalonate kinase (*PMVK*), *MVD*, isopentenyl-diphosphate delta isomerase (*IDI*) *1* and *2*, farnesyl diphosphate synthase (*FDPS*), and geranylgeranyl diphosphate synthase 1 (*GGPS1*) (*Kanehisa et al., 2012*) (*Figure 2*). A cycled primer extension and ligation-dependent amplification (CPELA) reaction for enrichment of target genomic

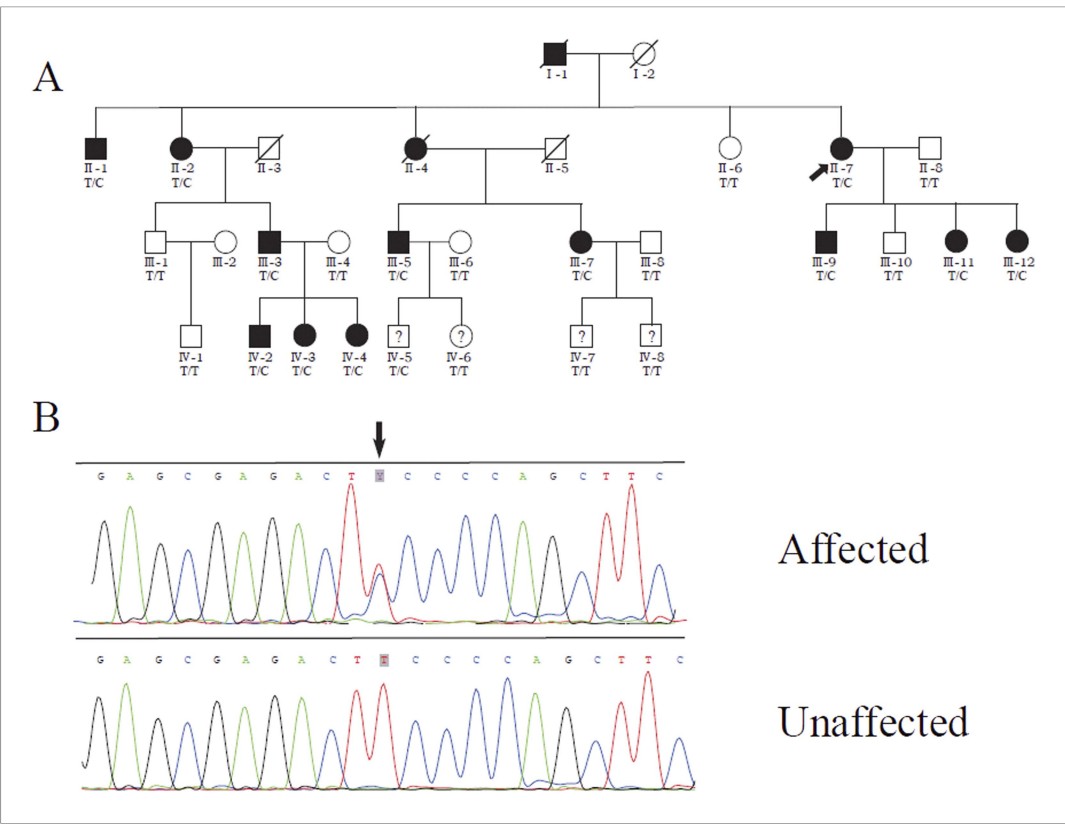

**Figure 1**. Identification of a *MVD* mutation in a porokeratosis (PK) family. (**A**) c.746T>C (p.Phe249Ser) in *MVD* displayed 100% co-separation with PK phenotype in this family (*Luan et al., 2011*). (**B**) Sanger sequencing chromatograms of proband (II-7, affected) and normal control (II-8, unaffected) at the c.746T>C mutation site indicated by arrow.

The following figure supplements are available for figure 1:

**Figure supplement 1**. Examples of six pedigree charts showing that each mutation displayed 100% co-segregation with the porokeratosis (PK) phenotype in the family.

**Figure supplement 2**. Two *MVD* mutations, c.302C>G (p.Pro101Arg) and c.683 G>A (p.Arg228Gln), for S-62 were located in the *trans* position because his daughter (S-62-D) carried only one of them.

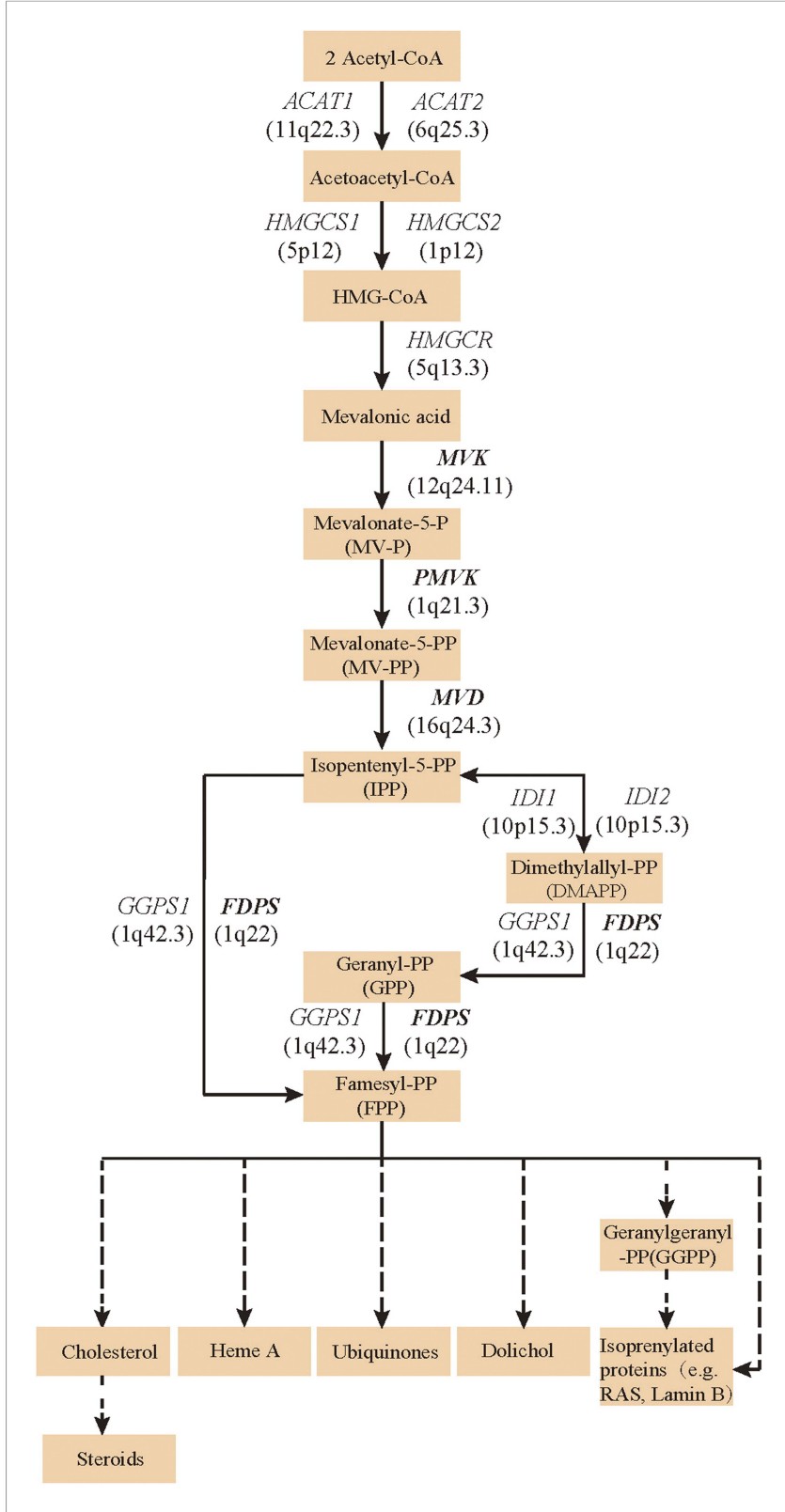

**Figure 2**. Isoprenoid biosynthesis via the mevalonate pathway. 12 member genes (*ACAT1, ACAT2, HMGCS1, HMGCS2, HMGCR, MVK, PMVK, MVD, IDI1, IDI2, FDPS, GGPS1*) were subject to mutation screening. The genomic loci of the 12 member genes are provided in parentheses. The illustration is adapted from the 00900 interactive map of the Kyoto Encyclopedia of genes and genomes (KEGG) (*Kanehisa et al., 2012*).

fragments and next generation sequencing (NGS) were performed in each sample. The NGS data were analyzed for rare single nucleotide variants (SNVs) with no record in dbSNP or a minor allele frequency of <1% in Chinese from 1000 genome database. After Sanger sequencing validation, 60 out of 61 SNVs called from NGS data were demonstrated to be true SNVs, which were predicted to impact protein function. Of these, 12 non-pathogenic SNVs were excluded from subsequent analysis (*Supplementary file 1*). In total, 48 mutation sites were identified in *MVK*, *PMVK*, *MVD*, and *FDPS* (*Figure 3*, *Figure 3—source data 1*).

## Large deletion mutations in *MVK* and *FDPS* were identified in PK patients

For 27 samples with no confirmed point mutation in *MVK*, *PMVK*, *MVD*, and *FDPS* genes, large deletion or duplication mutations were interrogated by analyzing the copy number of promoters and all exons in these genes. As a result, three deletion mutations were identified by the CNVplex assay. These three deletions were identified in four (F-26, F-40, S-69 and S-21), one (S-56) and one (F-6) cases, respectively. The breakpoints of three large deletions were also determined (*Figure 3*, *Figure 3—figure supplement 1*).

## Confirmation of genomic variations

We determined the genomic variations by Sanger sequencing in other family members apart from one with an unknown mutation. In 20 pedigrees with more than two blood samples collected, all patients carried the same mutation with the proband. On the other hand, no mutation was detected in any of the healthy family members. Except for one common *MVD* mutation of c.746T>C, all deleterious candidate SNVs and large deletions were undetected in healthy family members and 270 unrelated ethnically matched controls. In total, we identified 13 mutations in *MVD*, six mutations in *PMVK*, four mutations in *FDPS*, and 28 mutations in *MVK* in 113 of the 134 PK patients (*Supplementary file 2*). Notably, 21 of the 28 MVK mutations are novel (*Li and Zhang, 2014*). These mutations include 30 missense (58%), two start codon (4%), five nonsense (10%), seven indels (14%), four splicing defects (8%), and three large deletions (6%), and predictably affect functions of the respective enzymes. The two *MVD* mutations (c.746T>C and c.875A>G) were identified in 50 unrelated patients, accounting for 81% of all patients with *MVD* mutations. Among the 60 familial patients in whom at least one mutation was found, 30 had *MVD* mutations, three had *PMVK* mutations, two had *FDPS* mutations, 24 had *MVK* mutations, and one had both *MVK* and *MVD* mutations. Among the 53 sporadic patients in whom at least one mutation was found, 31 had *MVD* mutations, six had *PMVK* mutations, two had *FDPS* mutations, and 14 had *MVK* mutations. No mutation was found in the remaining 21 PK patients.

## Solute carrier family 17, member 9 gene (*SLC17A9*) mutation screening

For the 21 PK patients in whom no pathogenic mutation in *MVK*, *PMVK*, *MVD*, or *FDPS* was found, we sequenced all exons of the *SLC17A9* gene by Sanger sequencing. Mutations in the *SLC17A9* gene were recently reported to be another genetic cause for DSAP (*Cui et al., 2014*). However, no mutation in the *SLC17A9* gene was found in the 21 PK patients studied.

## Clinical manifestations in the 134 index patients with PK

To explore the genotype–phenotype correlations, we studied the clinical records of 134 index patients (92 males and 42 females) with PK. In all patients, the clinical diagnosis was supported by histopathological examinations. The onset of the disease ranged from birth to 78 years (mean age 31 years). The distribution of lesions could be categorized as localized and disseminated forms (*Schamroth et al., 1997*). Several classical variants were found to coexist in the same patient. However, no malignant degeneration was found in the lesions of these patients. Although all affected members in the same family carried the same mutation, each affected member showed different clinical manifestations and severity.

## Genotype–phenotype correlations in PK patients

We observed some interesting correlations between gene mutations and clinical phenotypes of PK (*Table 1* and *Figure 4*). First, giant plaque-type porokeratosis ptychotropica (PPt) with lesion diameters at least 5 cm appeared to be a unique phenotype associated with *MVK* mutations. This

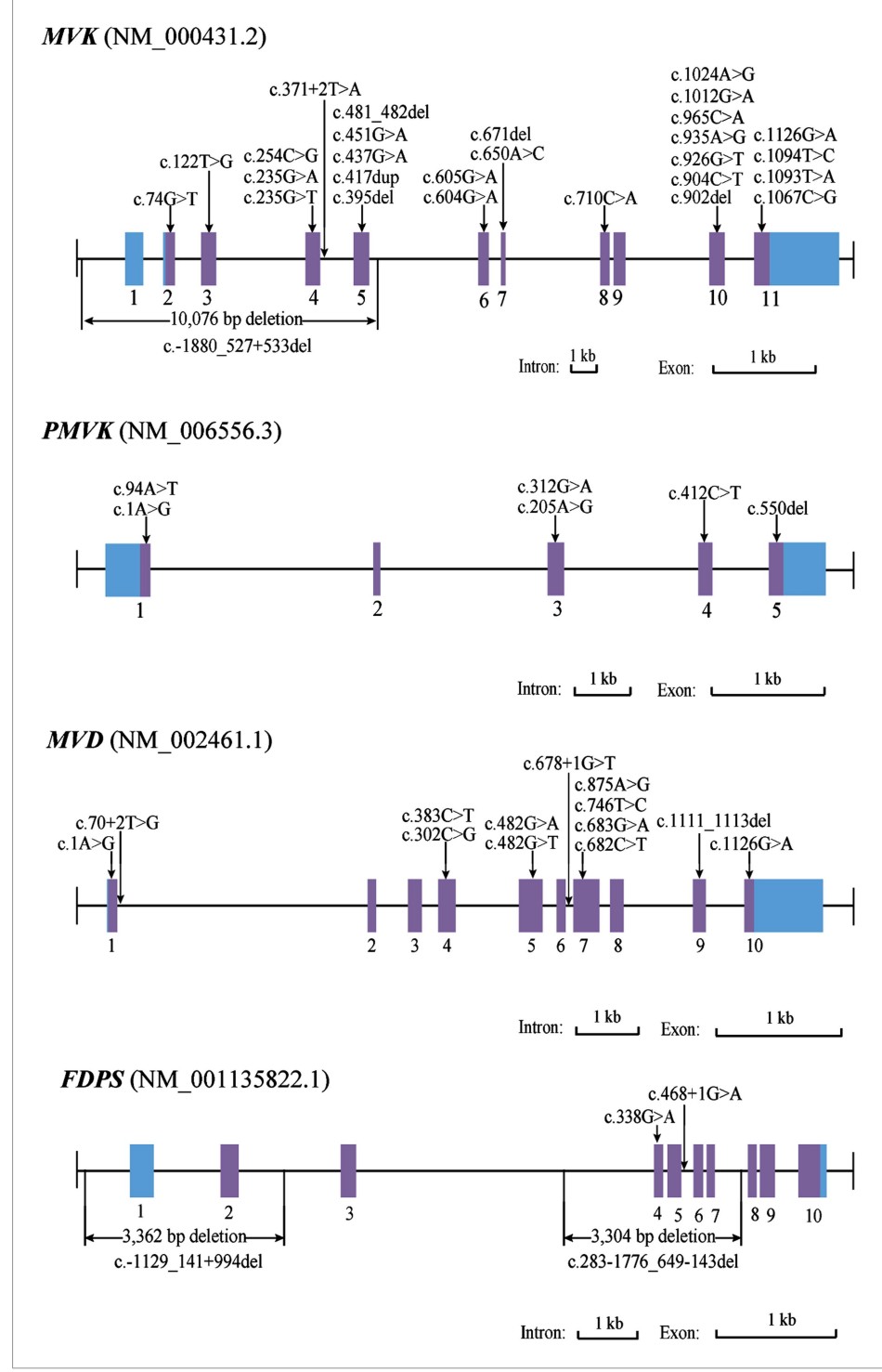

**Figure 3**. Mutational spectrum of *MVK*, *PMVK*, *MVD* and *FDPS* in 113 of the 134 porokeratosis (PK) patients.

The following source data and figure supplements are available for figure 3:

**Source data 1**. Sanger sequencing chromatograms of normal control and porokeratosis (PK) patients at 48 mutation sites in MVK, PMVK, MVD and FDPS.

**Figure supplement 1**. Breakpoint analysis for three large deletion mutations in *MVK* and *FDPS* genes.

*Figure 3. Continued*

**Figure supplement 2**. Illustration of amplification of multiple target DNA fragments mediated by cycled primer extension and ligation.

**Figure supplement 3**. The principle of CNVplex technology.

feature was observed in 50% (19/38) of index patients with *MVK* mutations, but not in any patients with *MVD*, *PMVK*, or *FDPS* mutations. In addition, patients with *MVK* mutations generally showed the widest range of phenotypes in terms of both the number and size of lesions. Second, localized genital PK (*Chen et al., 2006*) and porokeratoma (*Walsh et al., 2007*) appeared to be unique phenotypes associated with *PMVK* mutations. Third, in PK patients with *MVD* mutations, the age of onset spanned from 5 to 70 years, and the diameter of the lesions was generally less than 2 cm. Five females and one male with *MVD* mutations manifested mild solar facial PK (*Sharquie and Al-Baghdady, 2003*), which was not found in any patients with mutations in other genes. Fourth, in patients with *FDPS* mutations, the number of lesions was generally more than 500 and the diameter of the lesions was less than 1 cm.

As a whole, the lesions harboring mutations in *MVD* and *FDPS* genes tended to be more homogeneous and superficial than those carrying mutations in *MVK* and *PMVK* genes. The phenotypes of *MVD* and *FDPS* mutations on the natal cleft were mild and unobvious. However, some patients harboring *MVK* and *PMVK* mutations firstly presented with untypical lesions of genitogluteal porokeratosis (GGP).

## Reduced expression of wild allele and RNA editing in LTs

The expression of the wild allele in 10 out of 13 LTs was significantly reduced at the 1% significance level (*Figures 5–7*, *Supplementary file 3*). In contrast, the expression of the mutant allele was significantly reduced in almost all NNS (except one from F-42) carrying a mutation resulting in nonsense, frameshift, or splicing defect. As for LT from F-31, the mutant to wild allelic ratio in genomic DNA was >3 compared with the ratio of about 1 in the corresponding NNS genomic DNA. However, no deletion or duplication of all *MVK* exons was detected in F-31, which might indicate that gene conversion of the wild to the mutant allele occurred in the LT (*Figure 5*). Gene conversion might also occur in the thigh LT of F-43 (*Figure 7*). No copy number change or CpG island methylation was detected at the corresponding mutated genes in all LTs. A mutation of c.1003G>A (p.Gly335Ser) on the wild allele (T) of *MVK*: c.1093T>A was identified in the cDNA of the LT without AEI but not the cDNA of NNS collected from the left forearm of F-38 (*Figure 6*). Surprisingly, this mutation of c.1003G>A was not identified in the genomic DNA of this LT either, which indicated that G-to-A RNA editing occurred on the wild allele of *MVK*: c.1093T>A. No mutation was observed in cDNA samples from other tissues.

## Discussion

In this present study we were able to identify mutations in four genes (*MVK*, *PMVK*, *MVD*, and *FDPS*) underlying PK by massively parallel sequencing and copy number variation analysis in 134 index patients with PK. In the 12 PK families where the affected members span three to six generations, we found mutations only in *MVK* or *MVD* but not in *PMVK* or *FDPS* (*Figure 1*, *Figure 1—figure supplement 1*). We also observed the correlation of genotypes and several clinical manifestations in the PK patients (*Table 1*), which might be helpful to simplify the classification of PK under the guidance of genetic testing.

Interestingly, two patients were identified to carry two mutations: c.302C>G (p.Pro101Arg) and c.683G>A (p.Arg228Gln) in *MVD* for S-62, and c.746T>C (p.Phe249Ser) in *MVD* and c.235G>A (p.Asp79Asn) in *MVK* for F-28. According to the prediction programs of bioinformatics, the c.235G>A in *MVK* and c.302C>G in *MVD* were less conservative than the other mutation carried in F-28 and S-62, respectively. The two mutations in *MVD* for S-62 were located in the *trans* position, for his daughter carried only one of them (*Figure 1—figure supplement 2*). S-62 had late-onset PK at the age of 50 and his 30-year-old daughter had no lesions to date. According to our observation, index patients

Table 1. Clinical characteristics and genetic causes of 134 index patients with PK

| | Genetic causes of 134 index patients with PK | | | | |
| --- | --- | --- | --- | --- | --- |
| | MVK (39*) | PMVK (9) | MVD (62*) | FDPS (4) | Unknown (21) |
| Sex | | | | | |
| Male | 27 | 9 | 38 | 1 | 17 |
| Female | 12 | 0 | 24 | 3 | 4 |
| Number of lesions | | | | | |
| 0–10 | 4 | 4 | 0 | 0 | 6 |
| 10–100 | 20 | 5 | 13 | 0 | 5 |
| 100–500 | 5 | 0 | 39 | 0 | 5 |
| >500† | 10 | 0 | 10 | 4 | 5 |
| Diameter of lesions | | | | | |
| Minimum | 2 mm | 5 mm | 1 mm | 1 mm | 1 mm |
| Maximum | 20 cm | 5 cm | 2 cm | 1 cm | 2 cm |
| Age of onset | | | | | |
| At birth | 0 | 0 | 0 | 0 | 3 |
| 0–20 | 23 | 4 | 20 | 1 | 0 |
| 20–40 | 12 | 3 | 28 | 1 | 9 |
| 40–60 | 3 | 1 | 12 | 2 | 5 |
| >60 | 1 | 1 | 2 | 0 | 4 |
| Variants of PK | | | | | |
| DSAP/DSP | 26 | 0 | 56 | 4 | 10 |
| SFP | 0 | 0 | 6 | 0 | 0 |
| PM | 23 | 9 | 0 | 0 | 6 |
| HPM | 13 | 3 | 1 | 0 | 0 |
| Giant plaque of PPt | 19 | 0 | 0 | 0 | 0 |
| Genital PK (localized) | 0 | 4 | 0 | 0 | 0 |
| Porokeratoma | 0 | 5 | 0 | 0 | 0 |
| LP | 1 | 1 | 3 | 0 | 5 |
| Comorbidity | | | | | |
| Psoriasis vulgaris | 4 | 0 | 2 | 0 | 0 |

*One PK patient (proband of family-28, female), who has both the mutation c. 235G>A (MVK) and the mutation c. 746T>C (MVD), was included in both MVK and MVD groups.
†The number of lesions is more than 500.
DSAP, disseminated superficial actinic porokeratosis; DSP, disseminated superficial porokeratosis; HPM, hyperkeratotic porokeratosis, LP, linear porokeratosis, PK, porokeratosis; PM, porokeratosis of Mibelli, PPt, porokeratosis ptychotropica; SFP, solar facial porokeratosis.

F-28 and S-62 did not present with more severe phenotypes or earlier age of onset or any other unique clinical features.

In this study we did not detect pathogenic mutation in 21 patients: five cases with linear PK (LP), six cases with solitary plaque-type PK, and 10 cases with disseminated PK (only one had family history). There were three possible explanations: (1) these patients might have other non-classical or mosaic/ somatic mutations that could not be detected by current mutation screening methods; (2) these patients might have mutations in unknown genes; (3) these patients might be acquired ISIP without mutation. Recently, mutations in the SLC17A9 gene were reported to be another genetic cause for DSAP (Cui et al., 2014). We sequenced all exons of SLC17A9 in these 21 patients by Sanger sequencing but found no mutations in SLC17A9.

A    From F-42 proband with *MVK* mutation  (c.371+2T>A, p.Glu76Glyfs*9)

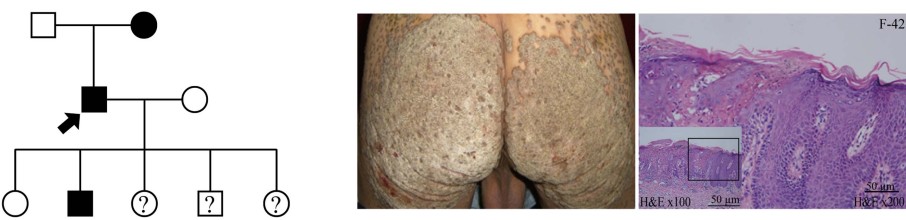

B   From F-60 proband with *PMVK* mutation  (c.412C>T, p.Arg138*)

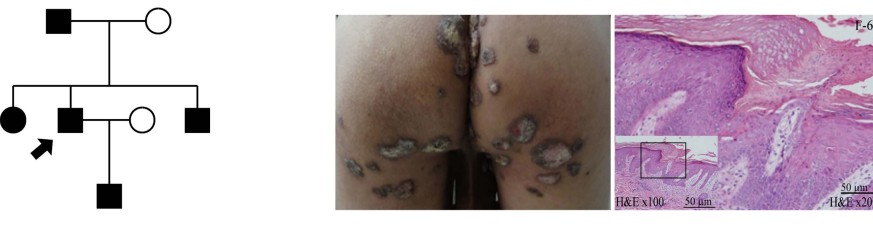

C    From F-36 proband with *MVD* mutation  (c.875A>G, p.Asn292Ser)

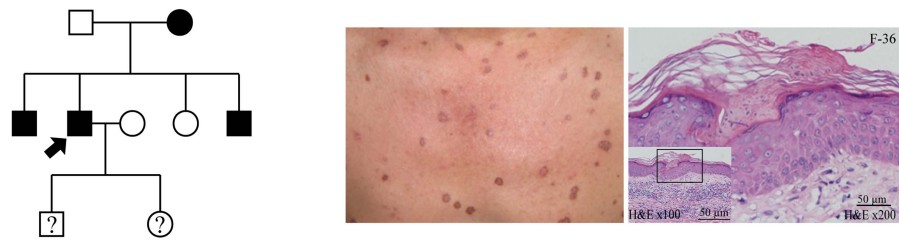

D    From F-47 proband with *FDPS* mutation  (c.338G>A, p.Arg113Gln)

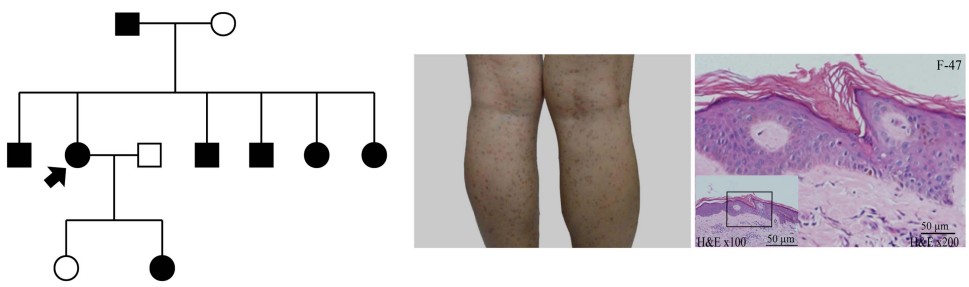

**Figure 4**. Representative clinical phenotypes and histopathology associated with the four genotypes. From left to right, pedigree charts, clinical phenotypes and the corresponding histopathology photos are shown correspondingly. (**A**) Family (F)-42 proband with *MVK* mutation showed giant hyperkeratotic plaque-type porokeratosis ptychotropica. (**B**) F-60 proband with *PMVK* mutation showed tumor-like porokeratoma in the genitogluteal region. (**C**) F-36 proband with *MVD* mutation showed discrete, red-brown annular keratotic papules or maculopapules on the chest. (**D**) F-47 proband with *FDPS* mutation showed multiple, small, superficial, annular papules with thread-like ridges on the legs. All histopathology showed cornoid lamella, a histological hallmark of porokeratosis with vertical columns of parakeratosis overlying an area of hypogranulosis with dyskeratotic cells.

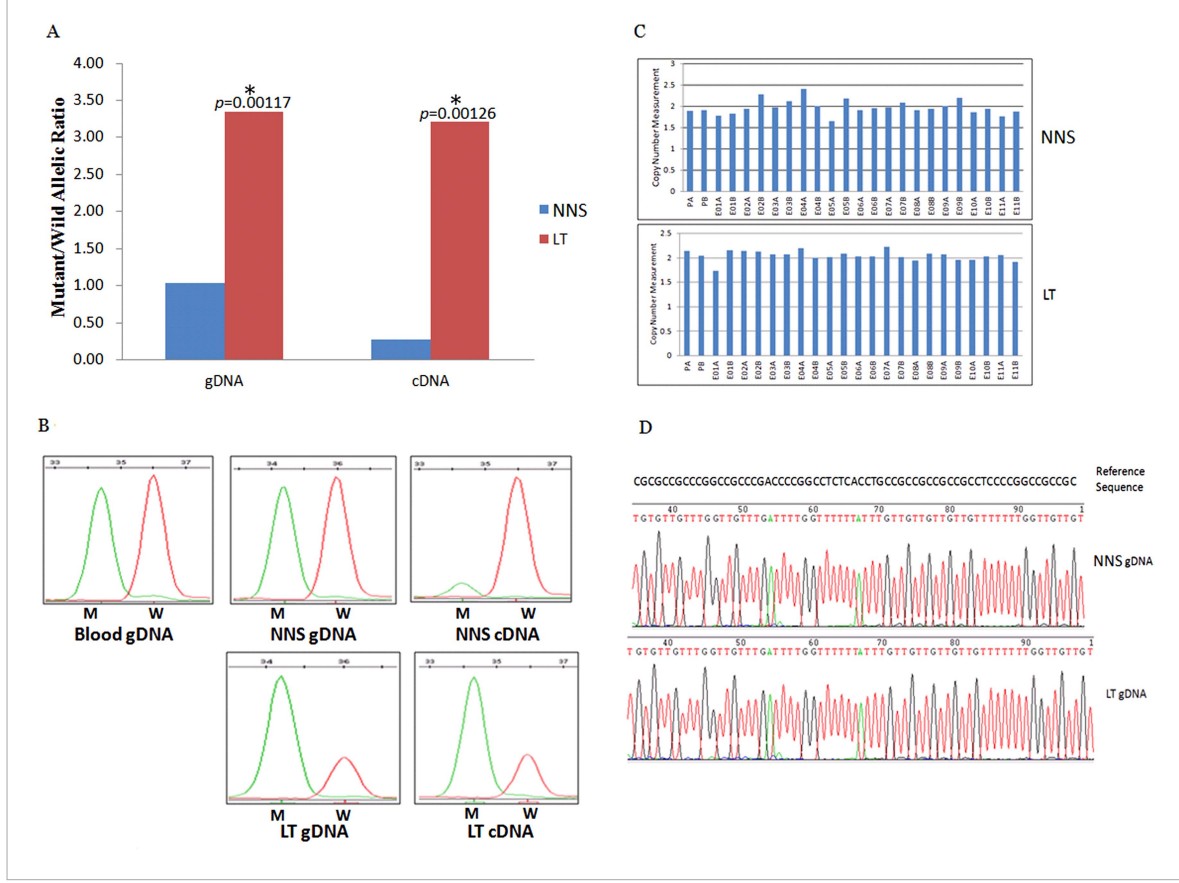

**Figure 5**. Gene conversion of the wild to mutant allele was identified in a buttock lesion from F-31 carrying a *MVK* mutation of c.395delT. (**A**) The mutant/ wild allelic ratios in genomic DNA (gDNA) and complementary DNA (cDNA) of lesional tissue (LT) and neighboring normal-appearing skin (NNS). The quantity of the mutant allele was about threefold and 10-fold more than the quantity of the wild allele in gDNA and cDNA, respectively. (**B**) The chromatograms of single nucleotide extension targeting the c.395delT mutation using the SNaPshot kit for five DNA samples (blood gDNA, NNS gDNA, LT gDNA, NNS cDNA and LT cDNA). The mutant peak was overpresented in both LT gDNA and LT cDNA. (**C**) No copy number change in genomic DNA of NNS and LT. PA and PB are probes in the promoter region, E01 to E11 designates exon 1 to exon 11, and A or B indicate two different probes in the same exon. (**D**) The bisulfite sequencing of NNS gDNA and LT gDNA. No methylation was observed for the targeted CpG sites in the promoter region of *MVK*.

RNA editing is a process of some discrete changes to specific nucleotide sequences of a RNA molecule which may be critical during normal development and diseases (*Witkin, et al., 2015*). G-to-A RNA editing in PK was first reported in our study. Interestingly, the RNA editing seemed to occur selectively on the one copy of *MVK* and this selective editing pattern seemed to be capable of being transmitted during mitosis. The mechanism underlying this observation should be investigated further.

Using AEI assays, we observed that the expression of wild allele was significantly reduced to various extents in most LTs (>77%). The variation in the expression reduction level might be due to the LT containing various amounts of normal-appearing keratinocytes and other different types of cells. Somatic loss of function of the wild allele in the epidermal stem cells might play a key role in the formation of lesions, and it might occur spontaneously and be induced by environmental factors such as ultraviolet light.

There was only one RNA editing identified in the LT, therefore the somatic loss of function of the wild allele in LTs of PK could be mainly caused by suppression of the wild allele through gene conversion in a minor proportion of cases or an unknown DNA methylation-independent epigenetic mechanism in most cases. Allelic imbalance of histone methylation modification unlinked to DNA

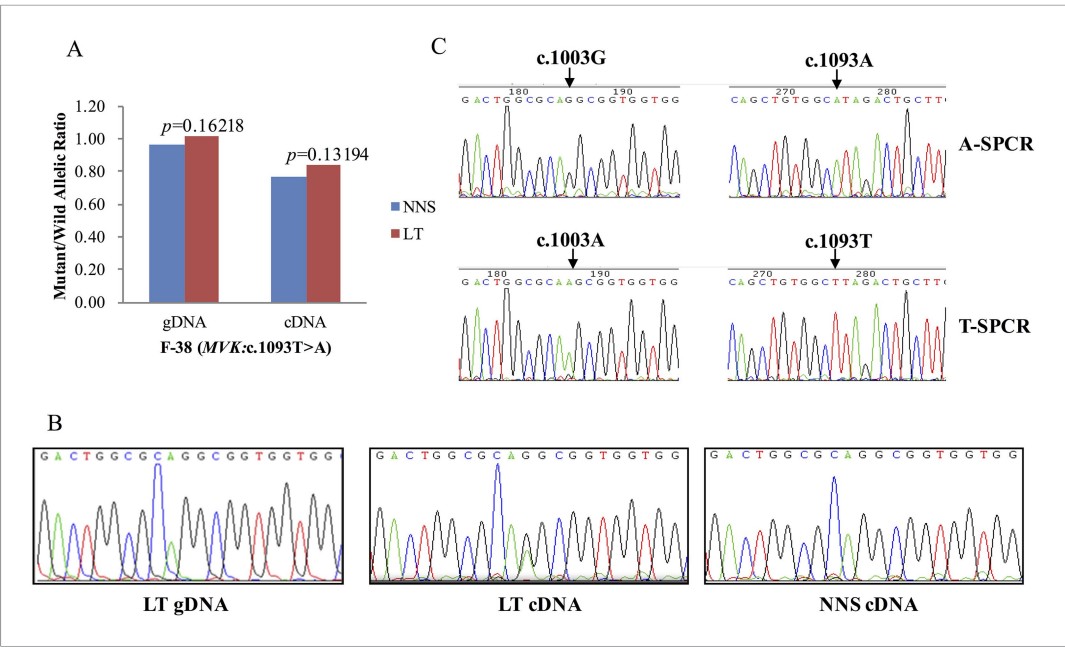

**Figure 6**. G-to-A RNA editing at position 1003 of the wild allele (T) of c.1093T>A in *MVK* was detected in a left forearm lesion from F-38. (**A**) No allelic expression imbalance was observed in lesional tissue (LT) from F-38. (**B**) A mutation of c.1003G>A was identified in LT cDNA, but not in neighboring normal-appearing skin (NNS) cDNA or LT gDNA. (**C**) Sequencing the c.1093A and c.1093T allele-specific PCR products indicated the mutant allele(A) of c.1003G>A was in the cis position with the wild allele(T) of c.1093T>A. A-SPCR, c.1093A–specific PCR; T-SPCR, c.1093T-specific PCR.

methylation, whose pattern is able to be transmitted during cell mitosis, could be the DNA methylation-independent epigenetic mechanism (*Xu et al., 2011*). Further investigation would be necessary to test this hypothesis. We did not find reduced expression or somatic mutation of the wild allele in only two LTs. The reason for this could be that the LT might contain many normal cells affecting the sensitivity of AEI and mutation detection.

As a starter button of PK, gene conversion of the wild allele to the mutant allele was first identified in this study. This observation is generally consistent with the phenomenon known as 'revertant mosaicism', in which the mutant allele is corrected in some stem cells. Revertant mosaicism is mainly identified in blood diseases such as Wiskott–Aldrich syndrome and Fanconi's anemia, and skin diseases such as ichthyosis and epidermolysis bullosa (*May, 2011*). Revertant mosaicism shows the bright side of gene conversion, but our observation revealed the opposite side. Therefore, gene conversion could be a natural 'rescue angel' in some recessive diseases or a 'damaging devil' in some dominant diseases, which could mostly affect tissues with extensive ability to self-renew for life.

Our findings support the notion that Mendelian diseases can be caused by multiple genes involved in the same metabolic or signal transduction pathway, or producing proteins on interaction. The four genes *MVK*, *PMVK*, *MVD*, and *FDPS* are members of the mevalonate pathway, suggesting that this pathway is involved in the pathogenesis of PK. The mevalonate pathway of isoprenoid biosynthesis provides precursors of isoprenoids, which are ubiquitous in living species and diverse in biological function (*Smit and Mushegian, 2000*) and serve as precursors of cholesterol, heme A, ubiquinones, dolichol, and isoprenylated proteins (e.g., RAS and Lamin B) (*Figure 2*). Moreover, it is also the subject of many pharmacological regulating drugs such as statins. It is well known that cholesterol is an important constituent of the cell membrane of most eukaryotic cells, and several inherited disorders have been linked to defects in cholesterol biosynthesis. The isoprenylated proteins play a role in the regulation of cell growth, division, and differentiation (*Moir and Spann, 2001*; *Zwerger et al., 2011*; *Davidson and Lammerding, 2014*) and are probably associated with the retained nuclei in the stratum corneum (i.e., parakeratosis). We propose that the accumulation of abnormal metabolites or

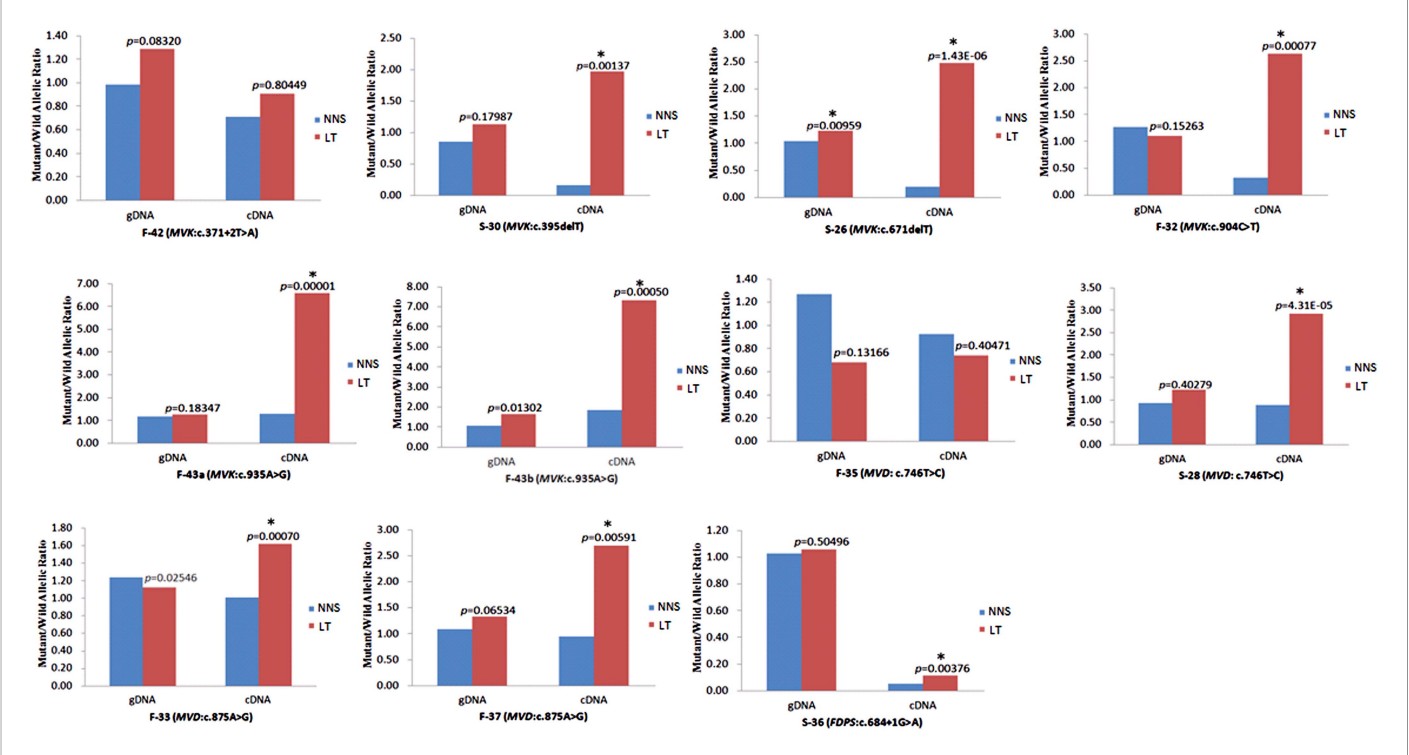

**Figure 7**. Significantly reduced expression of the wild allele in other nine lesion tissues. The Student t-test was performed to measure the difference in mutant/wild allelic ratios in genomic DNA (gDNA) or complementary DNA (cDNA) from lesion tissue (LT) and neighboring normal-appearing skin (NNS) for each mutation. The test score (p value) is presented above the LT bars. The asterisk (*) designates a significance level of 1%. F-43a and F-43b indicates the tissue sets of the left forearm and left thigh, respectively, from the same F-43 patient.

shortage of isoprenoids might predispose patients to idiopathic inflammation of the skin, and that correction of the abnormal isoprenoid biosynthesis might be a novel therapeutic direction for PK. Further functional studies will be necessary to confirm this hypothesis.

## Materials and methods

### Ethics statement
All procedures followed the guidelines of the Helsinki Declaration and were approved by the ethics committee and by the Scientific Ethical Committee of Fudan University. Participating centers provided local Institutional Review Board approval for genetic analysis. Study participants provided informed consent for genetic testing.

### Clinical samples
From 2001 to 2013, peripheral blood samples were collected from 134 index patients with PK (92 males and 42 females) and 180 family members (103 males and 77 females). Thirteen pairwise LTs and NNS were dissected from 12 index patients with a *MVK*, *MVD*, or *FDPS* mutation (*Supplementary file 3*). In addition, blood samples were collected from 270 healthy adult individuals undergoing routine medical examination in the hospital. The 134 index patients were diagnosed by at least two experienced dermatologists, based on both clinical features and histological examinations. In view of the family history, these index patients were divided into family (F) and sporadic (S) cases. Among the 61 familial cases, 40 index patients were probands from each family. We examined members of 21 families from different provinces of China, whose generations ranged from 2 to 6.

## DNA and RNA preparation

Genomic DNA was extracted from the peripheral blood using the QIAamp DNA blood mini kit (Qiagen, Germany). Total RNA was isolated from the skin using the RNeasy Protect Mini Kit (Qiagen, Germany) according to the manufacturer's instructions.

## Sanger sequencing and SNV genotyping

PCR products were sequenced by BigDye3.1 (Applied Biosystems, Foster City, CA, USA). Raw data were collected on an ABI 3130XL sequencer and mutations were called by the PolyPhred program (*Nickerson et al., 1997*). The 60 SNVs in 270 control samples were genotyped using the ABI PRISM SNaPshot Multiplex Kit (Applied Biosystems, Foster City, CA, USA) and ABI3730xl sequencer.

## CPELA

A total of 162 fragments covering the promoter region, 5′UTR, coding sequence and splicing site of 12 genes in the mevalonate pathway (*ACAT1*, *ACAT2*, *HMGCS1*, *HMGCS2*, *HMGCR*, *MVK*, *PMVK*, *MVD*, *IDI1*, *IDI2*, *FDPS*, and *GGPS1*) were amplified using the EasyTarget® amplification kit (Genesky Biotechnologies Inc, Shanghai, China) which was developed according to the CPELA method. The CPELA was a new method developed by Genesky Biotechnologies for fast and simple enrichment of multiple gene regions for massively parallel sequencing. The principle of CPELA is described in *Figure 3—figure supplement 2*.

## Massively parallel sequencing

The amplification products by CEPLA were mixed and size-separated by 2% agarose gel electrophoresis. Products of length 200–400bp were recovered. The final concentration of the library mixture was determined by real-time quantitative PCR and the average concentration for each library was estimated. The libraries tagged with different index sequences from several projects were well mixed in an appropriate concentration for each library corresponding to its aimed sequencing depth. The final library mixture was sequenced on MiSeq sequencer (Illumina) using Miseq reagent kit v2. The sequencing reads were separated for each sample by running CASAVA (Illumina Inc, San Diego, CA, USA) and target reads were determined by comparing them with fragment reference sequences (hg19) using the Blat program (*Kent, 2002*).

## Bioinformatics

In order to reduce the reading error from the sequencing reaction step, we first compared the two reads from the same cluster and integrated them into one correction read. The correction reads were then aligned to hg19 using the Burrows–Wheeler Aligner (BWA) (*Li and Durbin, 2010*). SNV calling was performed using both GATK (*McKenna et al., 2010*) and Varscan programs (*Koboldt et al., 2012*), and the called SNV data were then combined. The Annovar program was used for SNV annotation (*Wang et al., 2010*). The functional effect of non-synonymous SNVs was assessed by the PolyPhen-2, SIFT, and MutationTaster (*Ng and Henikoff, 2003*; *Adzhubei et al., 2010*; *Schwarz et al., 2010*). Non-synonymous SNVs with SIFT score of <0.05, Polyphen-2 score of >0.85 or MutationTaster score of >0.85 were considered as significant of not being benign. To sort potentially deleterious variants from benign polymorphisms, perl scripts were used to filter the SNVs against those of dbSNP135. Any SNV recorded in dbSNP135 and with a minor allele frequency of ≥1% in Chinese from 1000 genome database was considered as benign polymorphisms and therefore removed for subsequent analysis.

## CNVplex assay and breakpoint analysis

The copy number of the target regions was measured by a CNVplex assay, a high-throughput multiplex CNV analysis method recently developed by Genesky Biotechnologies. The principle of CNVplex technology is described in *Figure 3—figure supplement 3*. We utilized this technology for quantitative analysis of copy numbers of all 37 exons and upstream promoter regions in the *MVK*, *MVD*, *PMVK*, and *FDPS* genes for the blood DNA samples with no point mutation identified and the tissue DNA samples. Based on the copy number measurements for all target sequences, the breakpoints were estimated to be located between two neighboring probe target sites showing different copy numbers. Several primer sets flanking the two probe target sites were tested to amplify

the target region from case and control DNA samples using a long PCR protocol. Specific PCR products from case samples were sequenced using the ABI BigDye3.1 and the breakpoints were determined by blasting the sequences with human reference genome assembly.

## AEI assay

Both DNA and RNA were extracted from pairwise LT and NNS. Each RNA sample was reversely transcribed into cDNA twice using polyA and N9 primer mix and Reverse Transcriptase M-MLV (RNase H-) (New England Biolabs, England). For each pairwise tissue set, single nucleotide extension was used to quantitate the ratio of the mutant to the wild allele both in the tissue cDNA and DNA using the ABI PRISM SNaPshot Multiplex Kit (Applied Biosystems), followed by normalization to the ratio value in the corresponding patient's blood DNA as a reference of 1:1. As for cDNA from F-42 and S-36 patients with splicing defect mutations of *MVK*: c.371+2T>A and *FDPS*: c.684+1G>A respectively, fluorescent PCR followed by capillary electrophoresis was used to quantitate the ratios of mutant to wild transcripts.

## cDNA sequencing and bisulfite sequencing

Total RNA was reversely transcribed into cDNA. Two or three sets of primers were designed to amplify the cDNA of *MVK*, *MVD*, or *FDPS*, covering all coding sequences for each gene. The sequences of PCR products were determined by Sanger sequencing. Two allele-specific PCRs for c.1093T>A followed by Sanger sequencing were performed in order to determine whether the mutant allele (A) of c.1003G>A was in the cis position with the wild allele (T) of c.1093T>A. DNA from each pairwise LT and NNS were subject to bisulfite conversion using the EZ DNA Methylation Kit (Zymo Research Corporation, Irvine, CA, USA). PCR products amplifying the bisulfite converted DNA at CpG islands of *MVK*, *MVD*, or *FDPS* were sequenced on MiSeq sequencer (Illumina) using Miseq reagent kit v2 or on the ABI 3130xl genetic analyzer using the ABI BigDye3.1.

## Acknowledgements

We thank all patients and family members who participated in this study. This work is sponsored by grants from the Shanghai Pujiang Program, the Research Special Fund for Public Welfare Industry of Health, Guideline-oriented Research in the Management of Some Common and Severe Skin Diseases (No. 201002016), Construction of National Key Clinical Specialty in Shanghai, Ministry of Health. Special thanks should be given to Dr Lihua Zheng for helpful discussion.

## Additional information

### Funding

| Funder | Grant reference | Author |
|---|---|---|
| Shanghai Pujiang Program | none | Zhenghua Zhang |
| Research special fund for public welfare of health | 201002016 | Zhenghua Zhang |
| Ministry of Health of the People's Republic of China | Construct Construction of National Key Clinical Specialty in Shanghai, Ministry of Health | Zhenghua Zhang |

The funders had no role in study design, data collection and interpretation, or the decision to submit the work for publication.

### Author contributions

ZZ, Conception and design, Collection of patients, Acquisition of data, Analysis and interpretation of data, Drafting and revising the article; CL, ZJ, Conception and design, Acquisition of data, Analysis and interpretation of data, Drafting or revising the article; FW, JL, WL, SZ, JG, MF, XM, NS, XB, CG, ZZ, QH, LC, LX, JX, ZZ, Acquisition of data; RM, FY, LW, YL, WH, HP, Acquisition of data, Analysis and interpretation of data

### Ethics

Human subjects: All procedures followed the guidelines of the Helsinki Declaration and were approved by the ethics committee and by the Scientific Ethical Committee of Fudan University. Participating centers provided local Institutional Review Board approval for genetic analysis. Study participants provided informed consent for genetic testing.

## Additional files

### Supplementary files

• Supplementary file 1. 12 non-pathogenic rare missense or nonsense single nucleotide variants (SNVs) and major reasons for exclusion.

• Supplementary file 2. Characterization of 51 mutations identified in 113 of the 134 porokeratosis (PK) patients.

• Supplementary file 3. Mutant to wild allelic ratio measurements in both genomic DNA and complementary DNA of 13 pairwise tissue sets from 12 patients.

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
