## [Decision Letter]

Thank you for sending your work entitled “Genomic Variations of the Mevalonate Pathway in Porokeratosis” for consideration at *eLife*. Your article has been favorably evaluated by Stylianos Antonarakis (Senior editor) and three reviewers, one of whom, Helen Hobbs, is a member of our Board of Reviewing Editors.

The Reviewing editor and the other reviewers discussed their comments before we reached this decision, and the Reviewing editor has assembled the following comments to help you prepare a revised submission.

This paper explores the genetic basis of porokeratosis, a genetically heterogeneous autosomal dominant disorder of keratinization that includes both inherited and sporadic forms. Mutations in mevalonate kinase (MVK), a key enzyme in sterol synthesis, were shown previously by these authors to cause the disease in a subset of individuals, but the etiology in most affected individuals has remained unknown. Here the authors test the hypothesis that the disease is due to defective isoprenoid generation and function. First, they used linkage analysis followed by candidate gene sequencing to identify a second defective gene in the mevalonate pathway that causes this disorder, mevalonate decarboxylase (*MVD*). They then sequenced 12 genes in the mevalonate and isoprenoid pathways in 134 index cases and provide evidence that mutations inactivating 3 more genes in the pathway (*PMVK*, *MVD* and *FDPS*) also cause this disease. Collectively, implicated variants in the 4 genes identified explain disease in 98% of familial cases and 73% of apparently sporadic cases. They also provide evidence for correlation between the underlying gene and selected aspects of clinical presentation.

The three reviewers agree that the paper is clearly written and the major conclusions of the paper are supported by the data presented. Although this represents an important advance in the molecular characterization of this disorder, and further implicates and refines the disrupted pathway, the paper does not provide sufficient mechanistic insights to warrant publication in *eLife* and without such data is better suited for a subspecialty journal. How do defects in mevalonate metabolism cause the keratinization defects – toxic intermediary metabolites vs. deficiency of metabolic end-products? Most individuals were heterozygous for a mutation. Since the genes all encode enzymes, is there a mutation in the normal allele, or possibly in another allele in the pathway, in the skin lesions? Providing data to address any one of these questions would suffice.

Other major concerns:

1) While the authors describe that complete co-segregation between each putative mutation and disease was observed (and representative pedigrees are shown), these data should be expanded. While it might not be plausible to show pedigrees for all 60 familial cases with a defined mutation, it would be worth noting the size distribution of families for each gene and the number of families that achieved an independent LOD score >2.0 or >3.0, for example.

2) Curiously, it is stated that the vast majority of cases associated with variation in *PMVK* or *FDPS* were sporadic. If familial segregation was ever seen for these 2 genes, it should be shown. Genetic evidence that at least some of the sporadic mutations are de novo should be provided. It would also be informative to perform a formal burden test for these 2 genes to document enrichment for novel/rare and predicted deleterious variation in association with phenotype.

3) There is no evidence of pathogenicity for a number of the missense variants listed in Table 1 aside from the fact that they are present in cases and not controls (or 1 control, in the case of variant 44, *MVD* c.746T>C in Table 1). Providing information regarding the presence of these mutations in the public databases would be informative. A formal burden test for each new gene should be performed.

4) While the authors acknowledge that *MVK* mutations have previously been observed in other disorders, including mevalonic aciduria (MA) and hyper IgD syndrome (HIGDS), the same appears not to be true for the other 3 genes. Curiously, patients with biallelic mutations causing MA or HIGDS do not show porokeratosis, nor do their heterozygous parents. This is particularly puzzling because this group has identified porokeratosis in individuals heterozygous for some of the same mutations. Variation in environmental exposure (e.g. UV light) has been invoked as a potential mechanism, however this seems less tenable given the complete penetrance seen for porokeratosis in the described families. Somatic loss of the other *MVK* allele in skin lesions was previously excluded in a handful of patients, but this should be revisited in this manuscript.

---

## [Author Response]

*1) While the authors describe that complete co-segregation between each putative mutation and disease was observed (and representative pedigrees are shown), these data should be expanded. While it might not be plausible to show pedigrees for all 60 familial cases with a defined mutation, it would be worth noting the size distribution of families for each gene and the number of families that achieved an independent LOD score >2.0 or >3.0, for example*.

Among 61 familial cases, we examined members of 21 families from different provinces of China, whose generations ranged from 2 to 6. There were 5 core families (F-1, F-2, F-3, F-19 and F-24) in which more than 10 family members and blood samples were collected. The maximum two-point LOD score of F-1, F-2, F-3, F-19 and F-24 were 5.56 (D12S79), 8.62 (D12S84), 3.73 (D16S3074), 2.78 (D12S1583), 4.61 (D12S84), respectively. There were 40 probands from different PK families during the past 10 years, which could have been used for confirmation. All of them were adult and came to the out-patient department by themselves. We didn’t examine their family members and collected their blood samples. Therefore, it is difficult to calculate a reasonable LOD score in those non-core pedigrees. The sentence “We confirmed the genomic variations by Sanger sequencing in each family member. Each mutation displayed 100% cosegregation with PK phenotype in the family” may be misleading. Therefore, it was replaced with: “We determined the genomic variations by Sanger sequencing in other family members apart from one with unknown mutation. In 20 pedigrees with more than two blood samples collected, all patients carried the same mutation with the proband, on the contrary, no mutation was detected in all family healthy members”.

*2) Curiously, it is stated that the vast majority of cases associated with variation in* PMVK *or* FDPS *were sporadic. If familial segregation was ever seen for these 2 genes, it should be shown. Genetic evidence that at least some of the sporadic mutations are de novo should be provided. It would also be informative to perform a formal burden test for these 2 genes to document enrichment for novel/rare and predicted deleterious variation in association with phenotype*.

The sentence “In contrast, mutations in *PMVK* and *FDPS* were mainly found in proband or sporadic cases” was misleading. Since only 9 index patients were identified to carry *PMVK* mutations, and 3 of them were familial cases, and only 4 index patients were identified to carry *FDPS* mutations, and 2 of them were familial cases. Those index patients with negative family history were classified as the sporadic cases. Actually, the sporadic case might not be truly the only one patient in his family, because PK could be late-onset, undiscovered in non-exposure region. Therefore, we deleted this sentence in our revised manuscript. All 73 sporadic cases were adult and collected during the past 10 years, which could have been used for confirmation. Most of them lived separately with their parents and were out of touch with us for years. We didn’t examine those parents and get no informed consent and no peripheral blood from them. We are really sorry for that. The formal burden test is generally used for complex diseases with a large number of samples (Nat Genet. 2012, 29;44:623-630; Hum Mol Genet. 2012,21:R1-9). However, it is not suitable for our study.

*3) There is no evidence of pathogenicity for a number of the missense variants listed in*
Table 1
*aside from the fact that they are present in cases and not controls (or 1 control, in the case of variant 44,* MVD *c.746T>C in*
Table 1*). Providing information regarding the presence of these mutations in the public databases would be informative. A formal burden test for each new gene should be performed*.

The highest frequency mutation (c.746T>C in *MVD*) was identified in one control. This control was a 22 year-old male, who took routine medical examination in the hospital. He had no PK lesions till now and gave negative familial history of skin diseases. He could be predicted to develop PK in future. In view of its high frequency of occurence in PK, p-value of this mutation showed significant difference (p=0.00). Including this common mutation, there were 11 mutations had been identified in at least 2 index patients, but not in 270 controls. There are 7 mutations in *MVK* have been reported as indicated in Table 1, moreover, 5 mutations of them were identified in at least 2 index patients and 2 mutations of them were identified in 2 different sporadic cases. There were 4 mutations identified only in one family, but more than 2 patients in the pedigree were identified to carry this mutation and other healthy family members were not. Thus, at least 17 mutations in *MVK*, *MVD* and *PMVK* could be considered as pathogenic with strong evidence.

A formal burden test for each new gene may not be suitable for our study for the following two reasons: 1) the genetic and phenotypic information could provide solid support of *MVK, MVD* and *PMVK* being pathogenic genes of PK, especially *MVK* and *MVD*; 2) A formal burden test is generally used for complex traits with a large number of samples (Nat Genet. 2012, 29;44:623-630; Hum Mol Genet. 2012,21:R1-9). However, PK is regarded as monogenic disease with obvious dominant inheritance and a pedigree-based analysis could provide a more powerful way to identify the pathogenic genes, moreover, there were only 9 and 4 index patients with PK carrying different mutations in *PMVK* and *FDPS*, respectively.

*4) While the authors acknowledge that* MVK *mutations have previously been observed in other disorders, including mevalonic aciduria (MA) and hyper IgD syndrome (HIGDS), the same appears not to be true for the other 3 genes. Curiously, patients with biallelic mutations causing MA or HIGDS do not show porokeratosis, nor do their heterozygous parents. This is particularly puzzling because this group has identified porokeratosis in individuals heterozygous for some of the same mutations. Variation in environmental exposure (e.g. UV light) has been invoked as a potential mechanism, however this seems less tenable given the complete penetrance seen for porokeratosis in the described families. Somatic loss of the other* MVK *allele in skin lesions was previously excluded in a handful of patients, but this should be revisited in this manuscript*.

Mevalonate kinase deficiency syndrome (MKD) are early onset (usually <12 months), and it was reported that most hyper IgD syndrome (HIGDS) showed mild features of vasculitis (Arch Dermatol. 1994; 130: 59-65). Actually, it is a puzzle that why the patients with MA and HIGDS do not show PK. From the database of hereditary auto-inflammatory disorders mutations, c.417dupC and c.604G>A in *MVK* were both reported in HIDS and PK. As for those heterozygous parents, the possible reason was that pediatricians and rheumatologists might neglect to examine their skin, since they were not familiar with porokeratosis.

Somatic deletion of the wild *MVK* allele in skin lesions was not detected in our samples either, however, we observed the reduced expression of the wild allele in most skin lesions and a G-to-A RNA editing on the wild allele in one skin lesion. Therefore, the somatic loss of function, but not somatic deletion, of the wild allele in skin lesions should be common.